# An Ensemble of Transfer Learning Models for the Prediction of Skin Cancers with Conditional Generative Adversarial Networks

**DOI:** 10.3390/diagnostics12123145

**Published:** 2022-12-13

**Authors:** Amal Al-Rasheed, Amel Ksibi, Manel Ayadi, Abdullah I. A. Alzahrani, Mohammed Zakariah, Nada Ali Hakami

**Affiliations:** 1Department of Information Systems, College of Computer and Information Sciences, Princess Nourah bint Abdulrahman University, Riyadh 11671, Saudi Arabia; aaalrasheed@pnu.edu.sa (A.A.-R.); mfayadi@pnu.edu.sa (M.A.); 2Department of Computer Science, Al Quwaiiyah, Shaqra University, Shaqra 11961, Saudi Arabia; a.alzahrani@su.edu.sa; 3College of Computer and Information Science, King Saud University, Riyadh 11671, Saudi Arabia; mzakariah@ksu.edu.sa; 4Computer Science Department, College of Computer Science and Information Technology, Jazan University, Jazan 45142, Saudi Arabia; nmhakami@jazanu.edu.sa

**Keywords:** skin cancer, generative adversarial networks, conditional generative adversarial network, VGG16, ResNet50, ResNet101

## Abstract

Skin cancer is one of the most severe forms of the disease, and it can spread to other parts of the body if not detected early. Therefore, diagnosing and treating skin cancer patients at an early stage is crucial. Since a manual skin cancer diagnosis is both time-consuming and expensive, an incorrect diagnosis is made due to the high similarity between the various skin cancers. Improved categorization of multiclass skin cancers requires the development of automated diagnostic systems. Herein, we propose a fully automatic method for classifying several skin cancers by fine-tuning the deep learning models VGG16, ResNet50, and ResNet101. Prior to model creation, the training dataset should undergo data augmentation using traditional image transformation techniques and Generative Adversarial Networks (GANs) to prevent class imbalance issues that may lead to model overfitting. In this study, we investigate the feasibility of creating dermoscopic images that have a realistic appearance using Conditional Generative Adversarial Network (CGAN) techniques. Thereafter, the traditional augmentation methods are used to augment our existing training set to improve the performance of pre-trained deep models on the skin cancer classification task. This improved performance is then compared to the models developed using the unbalanced dataset. In addition, we formed an ensemble of finely tuned transfer learning models, which we trained on balanced and unbalanced datasets. These models were used to make predictions about the data. With appropriate data augmentation, the proposed models attained an accuracy of 92% for VGG16, 92% for ResNet50, and 92.25% for ResNet101, respectively. The ensemble of these models increased the accuracy to 93.5%. A comprehensive discussion on the performance of the models concluded that using this method possibly leads to enhanced performance in skin cancer categorization compared to the efforts made in the past.

## 1. Introduction

Skin cancer is a condition that develops when the DNA of healthy skin cells undergoes mutations that allow them to divide abnormally and turn malignant [1,2]. Excessive ultraviolet (UV) radiation exposure, time spent in the sun, and solarium usage are all possible causes [3]. Derived from the perspective of histology, skin cancer has an uneven cell structure with varying degrees of chromatin, nucleus, and cytoplasm [4]. Worldwide, skin cancer is one of the leading causes of death [5]. Basal Cell Carcinoma (BCC), Melanoma (MEL), non-melanoma skin cancer, and squamous cell carcinoma (SCC) are the most common types of skin cancer. Infrequent skin cancers, such as Kaposi Sarcoma (KS) and Actinic Keratosis (AK), include solar keratosis, lymphoma, and keratoacanthoma. Certain types of skin cancer are fatal and metastasis by nature. However, not all cancers are caused by malignant tumors. As the cancer of the skin begins in the epidermis, the outermost layer of skin, where it is visible to the human eye (National Cancer Institute, 2019), identifying a cancer as malignant (cancerous) or benign (non-cancerous) is frequently made based on a visual examination followed by a biopsy [6].

Melanoma is the most severe type of cancer since it is incurable. The majority of case turnover is death. Occasionally, melanoma develops from a cancer when its size, irritation, and hue are altered. Typically, non-melanomas are more prevalent than melanomas, yet melanoma is the leading cause of death from skin cancer [7]. However, early skin cancer detection and diagnosis will enhance the likelihood of recovery and survival; failure to achieve this will result in dire circumstances [8,9].

The pervasive and lethal character of the disease necessitates the development of an accurate, non-invasive diagnostic method. Most skin cancers are diagnosed using visual, clinical, and histological examinations. Frequently, medical diagnosis depends on the patient’s past, ethnicity, social behaviors, and sun exposure [10]. Visual inspection with the naked eye is typically incapable of identifying and revealing the intricacies. As a solution, dermatoscopy, an imaging tool [11] for skin cancer investigation, was developed. The optical dermatoscopy records dermatoscopic images using a high-resolution and magnifying camera lens. This recording technique eliminates the skin’s surface reflection, allowing for a real-time examination of the epidermis and dermis structures, in order that more visual data may be gathered from the deeper layers of skin, which will further aid in creating more precise Computer-Aided Diagnostic (CAD) systems. With only a visual examination, a dermatologist’s accuracy rate ranged between 65% and 80% [12]. However, dermatoscopy significantly improved the accuracy of early disease diagnosis [13]. A dermatologist’s eye examination and dermoscopic images have a combined accuracy rate from 75% to 84% [14,15].

Although dermoscopic images have improved accuracy, visual examinations still rely on the clinician’s expertise and subjective opinion to a large extent [16]. Color, dermal, contour, geometric, and texture features of cancers classify skin cancers. Skin cancers are difficult to classify visually. The degree of resemblance among the visual features of different cancer classes may lead to the incorrect recognition of cancers, especially when the cancer is in its early stages [17]. As a result, dermatologists frequently misclassify malignant and benign melanomas, which can devastate patients. It is more dangerous than the Squamous and Basalas since Melanoma spreads throughout the body significantly more quickly and attacks organs, including the brain [18] and liver [3], as well as Malware [19], cybersecurity [20,21,22], and mobile applications [23]. Dermatologists must develop new diagnostic techniques and methods to assist them in making early, accurate diagnoses of skin cancer to prevent or cure the disease due to the rapid development of skin cancer, the risk of metastasis, and the lack of therapeutic access [24]. New diagnostic instruments and methodologies are required for dermatologists and other medical professionals to accurately diagnose skin cancer.

Given the difficulty in diagnosing and treating skin cancer with the human eye, computer vision can be utilized for this purpose. To reduce the complexity of traditional Machine Learning techniques [25], a subject matter expert must first specify the features that will be employed. However, Deep Learning (DL) methods [26], a subfield of Machine Learning, can be trained on many benign and cancerous images. The DL model can determine whether a picture is malignant or benign by learning non-linear correlations. As a result, no domain expertise is required for feature extraction in DL. Using Convolutional Neural Networks (CNNs) for deep learning is the topic of this study.

The current work attempted to develop a novel diagnosis solution for skin cancer with an affordable computational cost and high accuracy, as early cancer detection is vital for both treatment and a cure for cancer. We develop an ensemble-based architecture that can be successfully employed to improve the accuracy of individual CNNs. The fusion of CNNs is one possible way to address the issues connected to the applicability of a single CNN for a given job. This is accomplished by allowing for additional classifiers, each based on a distinct CNN, to compensate for each other’s shortcomings. More specifically, we demonstrate how we can build a CNN ensemble to outperform the accuracy of individual neural networks trained on the available dataset. In addition, an investigation into the impact that data augmentation has on the overall performance of ensemble models was carried out. This study is the first of its kind in the early identification of skin cancer. Moreover, the presented Deep Learning models can be scalable to many devices, platforms, and operating systems, transforming them into contemporary medical instruments. The complete flow of this work can be visualized through Figure 1.

The contributions of the work are as follows:Exploration of image augmentation methods, such as flip, affine, linear contrast, multiply, and Gaussian blur (image transformation methods) to balance the dataset.Exploration of the Conditional GAN architecture for generating skin cancer images.Performance analysis of the fine-tuned pre-trained models VGG16, ResNet50, and ResNet101 on both balanced and unbalanced datasets.An ensemble algorithm by combining the predictions of the three fine-tuned models to improve the performance obtained by deep individual models.

The remainder of the manuscript is organized as follows. Section 2 discusses the previous research undertaken on the topic. Section 3 provides a complete mathematical explanation and visual results for the proposed methodology. In Section 4 and Section 5, the experimental design and findings are discussed. Finally, the conclusion is presented in Section 6.

## 2. Literature Review

We have conducted an extensive literature search to identify the contributions in skin cancer work among journals, such as Molecules, Nature, IEEE conferences and transactions, Springer, and Elsevier journals.

Several studies have utilized databases of dermoscopic skin cancers to aid in diagnosing cancers. Early studies on skin cancer mainly focused on various algorithms for categorizing skin cancers using traditional AI approaches, which typically begin with a phase of manual feature extraction, followed by a distinct period of classifier training. Early attempts to distinguish between skin cancers that were MEL or non-melanoma depended on low-level, manually-created characteristics [27]. Handcrafted features for dermoscopic images often have a low generalization power due to a lack of biological principles, understanding, and human intuition. Low-level handcrafted traits cannot distinguish complex skin cancers. In addition, there were considerable visual similarity challenges, high levels of intra-class disparity, and the appearance of artifacts in dermoscopic images that resulted in poor performance [28]. Therefore, deep learning and CNNs are unquestionably the preferred techniques in many computer vision applications [29,30,31].

The authors in this work [6] trained on a dataset of over 100,000 clinical images annotated by experienced dermatologists using Inception-v3 architecture. Deep CNN was developed for malignant melanomas against benign nevi and keratinocyte carcinomas versus benign seborrheic keratoses, which are two crucial binary classifications. The first involves identifying the most common cancers, and the second involves identifying the deadliest skin cancer. Differentiating benign nevi from malignant melanomas achieved 72.1 ± 0.9% accuracy, which was better than the dermatologist discrimination rates.

AlexNet was used by the authors of [32] to classify three different cancers (melanoma, common nevus, and atypical nevus). The ph2 dataset is used to train and test the proposed model [33]. The well-known quantitative measures of accuracy, sensitivity, specificity, and precision are used to evaluate the proposed method’s performance, with 98.61%, 98.33%, 98.93%, and 97.73% obtained, respectively.

An ensemble strategy for Convolutional Neural Networks (CNNs) has been suggested by the authors of [34], incorporating intra-architecture and inter-architecture network fusion in its design. Feature abstraction levels are represented by various CNN architectures in the proposed method. For each network, the in-depth features were used to train different Support Vector Machine (SVM) classifications. The proposed algorithm has an area under the Receiver Operating Characteristic (ROC) curve for melanoma classification of 87.3% and an area under the ROC curve for seborrheic keratosis classification of 95.5% when tested on the 600 test images from the ISIC 2017 skin cancer classification challenge.

Patch-based attention architecture suggested by the authors of [35] in 2020 provides a global context between high-resolution patches. Patch-based attention enhanced the mean sensitivity by 7% in the three pre-trained architectures studied.

The two-phase strategy presented by the authors of [36] includes mid-level features. The authors first identified the region of interest using dermoscopic images and then used pre-trained algorithms to extract information from the images. Their feature-based mid-level algorithm achieved a ROC of 0.87 for MEL and 0.97 for BKL.

In another work [37] explored how accurately skin cancer can be diagnosed due to the development of Convolutional Neural Networks (CNNs) [38]. This research demonstrates the skin cancer classification approach using the HAM10000 dataset. Implementation, training, and evaluation of VGG16, VGG19, and a Deep CNN are also proposed.

Similarly [39] used dermoscopic images from the MNIST HAM10000 dataset in this study. Along with DL, image augmentation techniques helped in boosting the total number of images. The authors turned to the transfer learning approach for the final boost in image classification precision. CNN’s weighted average accuracy of 0.88%, weighted recall average of 0.74%, and weighted F1 score of 0.77% were all achieved with our model. The ResNet model’s transfer learning method produced an accuracy of 90.51%.

A pre-trained DarkeNet19 deep neural network model was utilized by the authors of [24] to generate image gradients by tweaking the parameters of the third convolutional layer. Next, the High-Frequency and Multilayered Feed-Forward Neural Networks (HFaFFNN) are used to merge all visual images. Darknet-53 and NasNet Mobile are then used to train two deep models that can be finely tailored to the chosen datasets. Later, the idea of using transfer learning to train both models is investigated, with the input feed generating images of localized cancers. The collected characteristics are combined using the Parallel Max Entropy Correlation (PMEC) method in the next stage. An approach called Entropy-Kurtosis Controlled Whale Optimization (EKWO) is used to avoid overfitting and to select the most discriminating feature information. Three datasets, HAM10000, ISBI2018, and ISBI2019, were used in this study.

In most instances, a lack of data or an imbalance of data between classes included in the dataset is the fundamental cause of poor performance. A recent study [40] created a Deep Generative Adversarial Network (DGAN) multiclass classifier capable of generating skin disorder images by learning factual data distribution from publically available datasets. To deal with the class-imbalanced dataset, they used images from several internet databases [41,42]. Improving the DGAN model’s stability during training is a significant task. To analyze GAN’s performance, they created two CNN models based on ResNet50 and VGG16 and tested the models with labeled and unlabeled data. DGAN outperformed conventional data augmentation by 91.1% for unlabeled and 92.3% for labeled datasets. CNN models with data augmentation obtained 70.8% accuracy on unlabeled data.

## 3. Materials and Methods

### 3.1. Dataset

This study utilized the HAM10000 [43] dataset, which stands for “Human Against Machine with 10,000 training photos”. This dataset was used as the ISIC 2018 challenge training set (Task 3) [44]. To compile the collection, dermatoscopic photographs from diverse communities worldwide were used. Data collection was conducted to include all vital diagnostic categories linked with the field of pigmented cancers. Therefore, seven distinct skin cancer types were identified, Actinic Keratosis (AKIEC), Basal Cell Carcinoma (BCC), Benign Keratosis (BKL), Dermatofibroma (DF), Melanoma (MEL), Nevus (NV), and Vascular Cancer (VASC). The data collection contains 10,015 images, each of 600 × 450 pixels. Figure 2 depicts a selection of images representative of all groups of cancers.

A metadata file including demographic information for each cancer in question was supplied as supplementary data. In other instances, the gold standard is a follow-up examination, expert Consensus (confocal) or confirmation by in vivo confocal microscopy, with Histopathology (histo) accounting for more than half.

The objective of the current work was the classification of skin cancers. To expedite the development of the model, the images in the dataset were rescaled to 256 × 256 pixels. The three distinct datasets were generated by partitioning the original dataset into three sections, including train, validation, and test sets, each consisting of 70%, 10%, and 20% of the whole dataset’s images. The statistical breakdown for the seven different categories is presented in Table 1.

### 3.2. Data Augmentation

From Table 1, it was evident that the number of images for the seven classes spans a range of 115 (DF) to 6705 (NV). The HAM10000 dataset was unbalanced. The skewed dataset may cause overfitting while training the model [45]. To solve this data scarcity for some classes that will affect the classification model’s efficiency, the data augmentation method was employed. Data augmentation is a method that undergoes random transformations on the images to increase the count of images for underrepresented classes without the overhead of collecting more images [46]. The possibilities for image augmentation were enormous, such as rotation, translation, and flipping.

In the present work, we conducted several operations as shown in Table 2, which are all available on python library images [47]. After augmentation, the images are shown in Figure 3.

### 3.3. Image Generation Conditional Generative Adversarial Networks (CGANs)

In addition, we investigated the idea of generating synthetic data to solve the class imbalance issue. The well-known CGAN [48] architecture is used to create the images. Figure 4 depicts the high-level design of the network.

Generative Adversarial Networks (GANs) [49] normally use a generator to learn how to create new images and a discriminator to learn how to distinguish between artificial and genuine images. However, there was no mechanism to regulate the images generated, such as the development of multiclass data.

A conditional setting governs the training of the generator and discriminator in CGANs (such as class labels or data). Discrimination judgments will be based on the generated images and their labels, with the former being more important to the discriminator. The ideal model can learn multimodal input-to-output mapping by being fed a range of contextual inputs. Following the creation of the model, we retained only the trained generator model that was utilized to generate the synthetic images. Some of the synthetic images generated by the trained CGAN model are displayed in Figure 5.

### 3.4. Classification Model Development

The majority of real-world datasets suffer from data insufficiency issues and constructing the most effective deep learning model for computer vision applications necessitates a large number of data. In addition, there will be insufficient processing capability if the dataset is enormous. With the development of the transfer learning [50] approach, these issues were resolved. Transfer learning is the most extensively utilized method for categorization tasks. As an alternative to training from scratch, it is a popular strategy in deep learning where pre-trained models are employed as the starting point. It is common practice to utilize deep models, such as VGGNet, ResNet, etc., which have been pre-trained for a large and challenging image classification task, such as ImageNet 1000-class. The feature extractors of these deep models will be crucial for capturing critical features for classification. Only the dense layers at the output must be modified in accordance with the number of classes we wish to create.

In the present work, we used three deep learning models, VGG16 [51], ResNet50, and ResNet101 [45], that were all pre-trained on the ImageNet dataset. The designed architecture is shown in Figure 6.

### 3.5. Experimental Setup

The quantity of hyperparameters rises as the CNN depth increases. The CNN model has a large number of hyperparameters [52], including the number of layers, the number of kernels in each layer, the size of the dense layer, the learning rate, the number of convolutions, the number of max-pooling, the number of dense layers, the number of epochs, the batch size, the optimizer, the activation function, etc. The selection of these hyperparameters is typically carried out manually and involves expensive trial and error. As a result, CNN training typically takes a long time since different hyperparameter configurations are evaluated in different rounds. In the interim, new CNNs typically have an accelerated increase in the number of layers together with hyperparameters. Therefore, trying to manually determine the almost ideal hyperparameter setup for a CNN at a reasonable cost is virtually pointless. On the other hand, effective hyperparameter adjustment can improve the CNN model’s overall performance. Due to this goal, many researchers have turned to the tuning of CNN hyperparameters as an optimization problem. These hyperparameters have a significant impact on the CNN model’s result. The near-optimal hyperparameter setup for a CNN cannot be manually determined by cost-effectively examining all conceivable combinations. Therefore, correct hyperparameter tuning of CNN is designed as an optimization problem with the goal of improving the CNN model’s overall performance.

### 3.6. VGG16

In VGG16 CNN, there are 16 layers with an input dimension of 2,242,243 to the network. In fact, VGG16 stands out from other implementations since it uses a 2 × 2 stride 2 filter with the same padding and max-pooling layer, rather than occupying many hyperparameters. Throughout the architecture, the convolution and max-pooling layers are arranged in the same manner. For output, there are two Fully Connected (FC) layers and a SoftMax layer.

The 16-layer network used by VGG16 is significantly deeper than AlexNet but provides a simpler network by substituting numerous 3 × 3 kernel-sized filters for a single large kernel-sized filter. ImageNet, a collection of more than 14 million photos divided into 1000 groups, scored 92.7% on the top five test.

The ReLU from AlexNet is carried out by 13 convolutional layers and 3 fully linked layers of VGG16. The fully connected layers are utilized for classification, whereas the convolutional layers are used to extract features. Each of the 64-feature kernel filters in the first and second convolutional layers is 3 × 3 in size. As the input image (a RGB image with depth of 3) moves through the first and second convolutional layers, its dimensions are modified to 224 × 224 × 64. The output is subsequently transferred with a stride of 2 to the max-pooling layer. With a filter size of 3 × 3, the third and fourth convolutional layers are composed of 128-feature kernel filters. After these first two layers are applied, a max-pooling layer with stride 2 is used to compress the output to 56 × 56 × 128 pixels. The fifth, sixth, and seventh layers are convolutional layers with a kernel size of 3 × 3. All three of these layers combine to employ 256 feature maps. A max-pooling layer with stride 2 is placed after these layers. With kernel sizes of 3 × 3, there are two sets of convolutional layers from the eighth to the thirteenth. Each of these convolutional layers has 512 kernel filters. A max-pooling layer with a stride of 2 follows these layers. The sixteenth layer, which has 1000 neurons, is a SoftMax output layer for classifying 1000 photos from the ImageNet dataset. The fourteenth and fifteenth layers are fully linked hidden layers with 4096 neurons each. VGG16 is utilized as a feature extractor and classifier in image processing applications notwithstanding its utility in image processing and classification. The GPU used in this implementation is GPU DELL EMC 740, and the RAM is 128 GB. Moreover, the programming language used is Python, and the IDE is Jupyter notebook environment on Google Colab.

### 3.7. ResNet50

Deep learning research has seen a widespread trend toward increasing the number of layers in CNN architecture in order to improve performance. However, there was a vanishing/exploding gradient problem as the layers increased. Therefore, the concept of “Residual Network” was introduced in architecture. When the network uses the “skip connections” idea, some subsequent connections are skipped, and the output is directly connected. The ResNet variation that contains 50 layers is called ResNet50.

Therefore, this combination of algorithm hyperparameters (i.e., optimizer of sgdm, Mini BatchSize of 40, MaxEpochs of 5, and Initial LearnRate of 10–4) was selected for ResNet50 and ResNet101 transfer learning.

### 3.8. ResNet101

The 101 layered ResNet is ResNet101. The architecture is more complex than ResNet50 as it contains more trainable parameters.

### 3.9. Ensemble Algorithm

Three deep learning models were trained here for skin cancer prediction. However, we knew that a single algorithm might not provide the most accurate forecast for a specific dataset. There were limitations to machine learning methods and developing a model with great accuracy is difficult. By combining multiple models, overall accuracy could be improved. The combination can be performed by averaging the output of each model with two goals in mind: Minimizing model error and preserving its generalizability. Each model predicted the likelihood of each class’s forecast given its class. Taking the average of the prediction probabilities by the three models may result in a performance increase. The architecture for the ensemble algorithm is shown in Figure 7.

### 3.10. Performance Evaluation

The suggested model architecture was evaluated for its performance in predicting skin cancers using many performance assessment indicators. Accuracy, recall, precision, and F1 score are the four measures. True Positives (TP) and False Negatives (FN) are the numbers of positive images accurately predicted. In contrast, the number of incorrectly anticipated negative images is known as False Positives (FP), while the number of accurately predicted negative images is known as True Negatives (TN) [53].

Accuracy: The ratio of the number of classes a model successfully predicts to the total number of predictions.

(1)
$$\mathrm{Accuracy}=\left(TP+TN\right)/\left(TP+TN+FP+FN\right)$$


Precision: Precision is defined as the proportion of the number of correct predictions divided by the total number of positive class predictions. Equation (1) is used to calculate Precision.

Recall: Recall is defined as the proportion of correct predictions divided by the number of actual counts of the positive class in the dataset. Equation (2) is used to calculate Recall.

(2)
$$\mathrm{Recall}=\left(TP\right)\;/\;\left(TP+FN\right)$$


F1 score: The F1 score represents the balance between Precision and Recall.

(3)
$$F1\;\mathrm{score}=\left(2\times \;Precision\times \;Recall\right)\;/\;\left(Precision\;Recall\right)$$


## 4. Results

### 4.1. Transfer Learning Model with the Unbalanced Dataset

Following the conclusion of the training for the VGG16, ResNet50, and ResNet101 architectures, predictions were made on the test set to understand how well they performed. The final layer of our design is called SoftMax, and this layer was responsible for producing the prediction probability of each of the seven classes. It is necessary to be aware of each model’s performance before selecting the most appropriate model for the classification of skin cancers.

An analysis was carried out on the data obtained from the VGG16, ResNet50, and ResNet101 models’ respective experiments. Figure 8 depicts the validation accuracy, error rate, and loss plots for each of the three models. It is possible to see that the accuracy has improved while the loss has decreased. Therefore, there was no evidence suggesting that any models would overfit.

The confusion matrices on the test predictions are shown in Figure 9, indicating the number of correct and incorrect predictions based on each class in the test dataset (AKIEC, BCC, BKL, DF, MEL, NV, VASC) on the test set. The performance evaluation metrics of the three models are shown in Table 3.

#### The Effect of the Ensemble Algorithm

The ensemble model combines the mean of the predictions by all three models. The confusion matrix of the ensemble model is shown in Figure 10. The performance analysis of the ensemble model on the unbalanced dataset is shown in Table 3. Finally, the class-wise performance is shown in Table 4.

### 4.2. Transfer Learning Model on the Balanced Data Obtained by Data Augmentation

Several image augmentation techniques were used to increase the number of images within the skin cancers. The images of skin cancer type “NV” were excluded from the data augmentation process since it was an overrepresented class. Only underrepresented groups underwent the image augmentation procedure. In addition, we attempted to create synthetic images for each class. However, as shown in Figure 5, the produced images did not appear to possess the characteristics that differentiate the seven types of skin cancers. We anticipate that these data will not be suitable for training the model for optimal performance. Therefore, the resulting CGAN data were excluded from the training data through augmentation. The performance metrics of the developed models VGG16, ResNet50, and ResNet101 are shown in Table 5.

### 4.3. The Effect of the Ensemble Algorithm

The performance analysis of the ensemble model on the balanced dataset is shown in Table 5, while the class-wise performance is shown in Table 4.

## 5. Discussion

A wide variety of categorizations exist for skin cancers, some of which were cancerous while others were benign. It is essential to determine the specific type of skin cancer, in order to determine whether the condition may progress to cancer and to ensure that the appropriate therapy is administered. Obtaining a cancer diagnosis at an earlier stage is essential if one wants to experience a full recovery from the disease. A delayed detection could cause the problem to become more complicated, putting a person’s life in danger.

It has been discovered that deep learning is the most effective way for determining the different types of skin cancers. However, to extract the appropriate features for identifying the various classes without the participation of a human, a massive amount of sampled data, acquired from patients in each class, are required [46]. However, the collection of this massive volume of labeled data is nearly impossible, particularly in the field of medicine. Consequently, the majority of the medical datasets that are accessible to the public have a problem with data imbalance [47]. As can be observed in Table 1, the dataset that was used for this investigation has some serious imbalances. Nevus, a type of benign cancer, accounts for the majority of the images and has an abundance in comparison to other categories. If you train the model using this type of dataset, there is a possibility that it will have a bias for the category that has the most images. The classification of skin cancers with a relatively low number of available examples for training resulted in an incorrect identification as the NV class. A similar thing can be seen from the confusion matrices in Figure 9, which show that the class Dermatofibroma (DF), which has the fewest number of images, is most commonly misdiagnosed as nevus.

Data augmentation was the most straightforward response to this problem. However, data augmentation might be one of two different types: Picture transformations using the conventional methods or the development of synthetic images using the capabilities of GAN architecture. Several of the earlier researchers successfully implemented the GAN algorithm as a means of image augmentation, and they obtained performance improvements as a result [40,48]. However, the synthetic images that were produced by our CGAN generator and displayed in Figure 5 did not appear to be sufficiently effective to contribute to the improved classification performance. Although prior work successfully applied the CGAN to produce synthetic skin cancer images using the HAM10000 dataset, training with a significantly better dataset has the potential to assist in improving the proposed model’s overall performance. However, the CGAN architecture that we designed was unable to generate images that were similar to genuine images. In fact, even humans could tell that these images had been artificially created by looking at them. The model could only produce the pink hue presented in the actual image and it was evident that they were unable to generate skin cancer images that had any distinctive characteristics for each class.

However, the conventional image enhancement was successful in accomplishing the goal of achieving the performance increase in all three models that were built. When using the balanced dataset, all of the performance evaluation measures show a higher value, which was achieved by having less instances of incorrect classifications. The ensemble model was another clever strategy that was utilized to obtain a higher level of performance. If the predictions from all three models are combined, there is a possibility that the overall performance will be improved. Moreover, it is possible that some models will not be able to identify certain classes, and the probability of prediction of those classes might be low. However, taking the average of the predictions of the three models might help in increasing the probability of correctly classifying those classes, which would help in reducing the number of false negatives and false positives. Within the scope of this study, the combination of all three models unequivocally demonstrated an increase in performance (Table 3 and Table 5). Furthermore, incorporating an ensemble model that was trained on the balanced dataset may improve the performance of virtually all classes (Table 4).

### Performance Comparison with Previous Works 

A large number of previous works were carried out on the HAM10000 dataset as a classification of skin cancers. Ref. [54] developed a MobileNet model for classifying the skin cancers. However, the overall achieved accuracy of the classification was 83.1%. The authors of [55] followed the transfer learning approach for efficient feature extraction using the ResNet50 and ResNet101 pre-trained models. The feature selection process was carried out from the large number of extracted deep features. The selected features were provided by the SVM and Radial Basis Function (RBF) for classification. However, the developed model could only achieve a performance of 89.8% even with the deep feature extraction and feature selection. Moreover, the authors of [56] used MobileNet for the same task. The model’s performance was carried out for improvement by upscaling and augmentation methods, and the researchers succeeded in achieving the task. However, the improved accuracy was limited to 83.23%. Furthermore, the authors of [57] achieved 85.8% on the HAM10000 dataset, whereas the authors of [58] enhanced the images by local color-controlled histogram intensity values before training the CNN model. The developed model could achieve an accuracy of 90.67%. Table 6 shows all the comparisons with respect to our work.

The currently developed model could achieve greater results, but the dataset suffered from an imbalanced data problem, which had a direct impact on the performance of the model developed from the dataset. With the typical data augmentation technique, performance has been enhanced, but certain cancer classes in the dataset continue to suffer from poor detection, particularly melanoma, a particularly serious form of skin cancer. In order to obtain high performance for the low-performing classes in the current study, a future study will integrate a more extensive dataset derived from various publically available skin disease datasets. In addition, a study of computation time vs. accuracy will be conducted to see how the model could be implemented in real-time, low-power medical devices.

## 6. Limitations

There were limitations to machine learning methods and developing a model with great accuracy is difficult. Additionally, combining multiple models is a large task and improving the accuracy is very difficult. Therefore, the combination can be carried out by averaging the output of each model with two goals in mind: Minimizing model error and preserving its generalizability.

## 7. Conclusions

The cancer that affects the skin is one of the deadliest forms of the disease. In addition, identifying it using methods, such as dermatoscopy and the naked eye, was time-consuming. Early identification of skin cancer may allow for a treatment that prevents them from progressing to more fatal forms. This work aimed to develop an effective deep learning model that could be used for the early diagnosis of seven different types of skin cancers. However, the process was not simple since the HAM10000 dataset that was used for this work appeared to be unbalanced. In order to find a solution to this problem, we investigated data augmentation methods, conventional image translations, and image generation possibilities. However, the work showed that the GAN architecture was not properly trained to generate the appropriate authentic-looking skin cancer images that qualify as the model training input. This was revealed by the GAN architecture which failed to generate these images. On the other hand, the image transformations might be able to produce a magnificent dataset to solve the problem of image imbalance. Moreover, an ensemble model, consisting of the VGG16, ResNet50, and ResNet101 models, that had been trained on both balanced and unbalanced datasets was developed, and its performance was analyzed. According to the findings of the study, an ensemble model that had been trained on a balanced dataset was able to produce the best results for skin cancer classification while also displaying less bias toward the category that contained the most significant number of examples (Nevus). The accuracy of the model, which was obtained as 93.5%, was significantly better than many of the previous efforts that had been made on the same dataset.

## Figures and Tables

**Figure 1 diagnostics-12-03145-f001:**
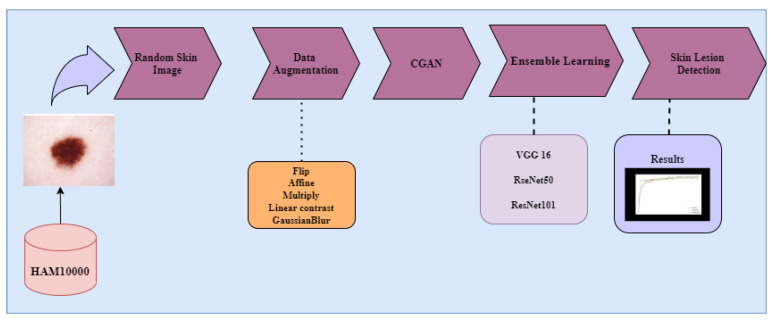
The complete flow of the work.

**Figure 2 diagnostics-12-03145-f002:**
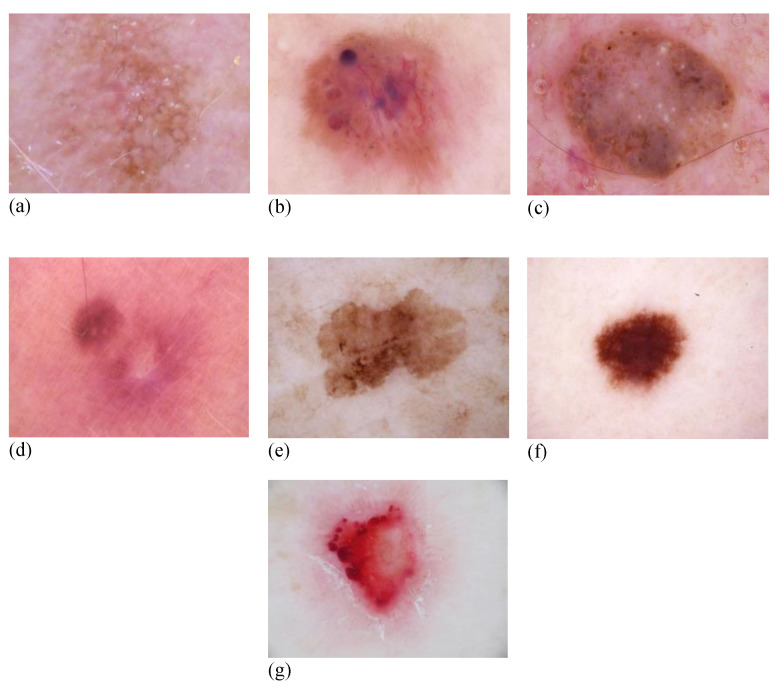
Sample skin cancer images from the dataset. (**a**) AKIEC; (**b**) BCC; (**c**) BKL; (**d**) DF; (**e**) MEL; (**f**) NV; and (**g**) VASC.

**Figure 3 diagnostics-12-03145-f003:**
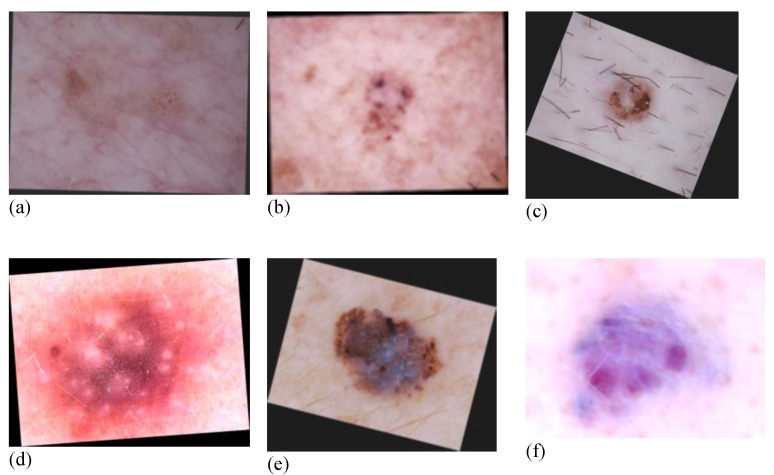
Augmented images. (**a**) AKIEC; (**b**) BCC; (**c**) BKL; (**d**) DF; (**e**) MEL; and (**f**) VASC.

**Figure 4 diagnostics-12-03145-f004:**
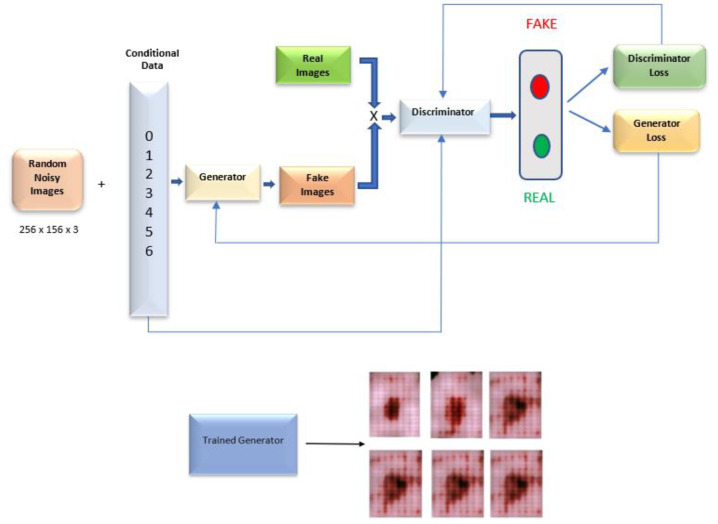
CGAN architecture.

**Figure 5 diagnostics-12-03145-f005:**
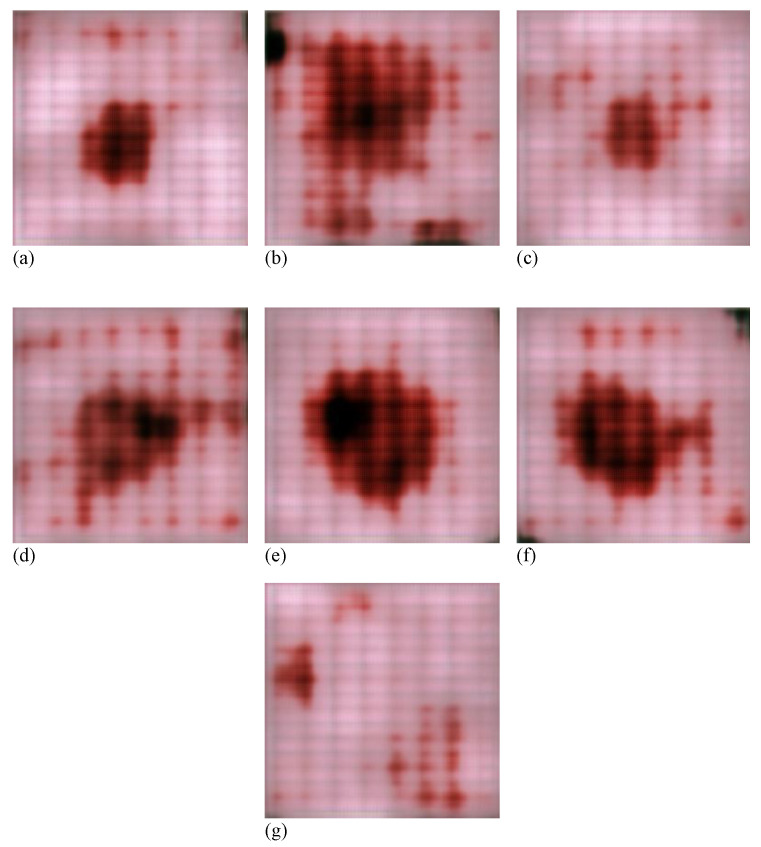
Some of the synthetic images from CGAN. (**a**) AKIEC; (**b**) BCC; (**c**) BKL; (**d**) DF; (**e**) MEL; (**f**) NV; and (**g**) VASC.

**Figure 6 diagnostics-12-03145-f006:**
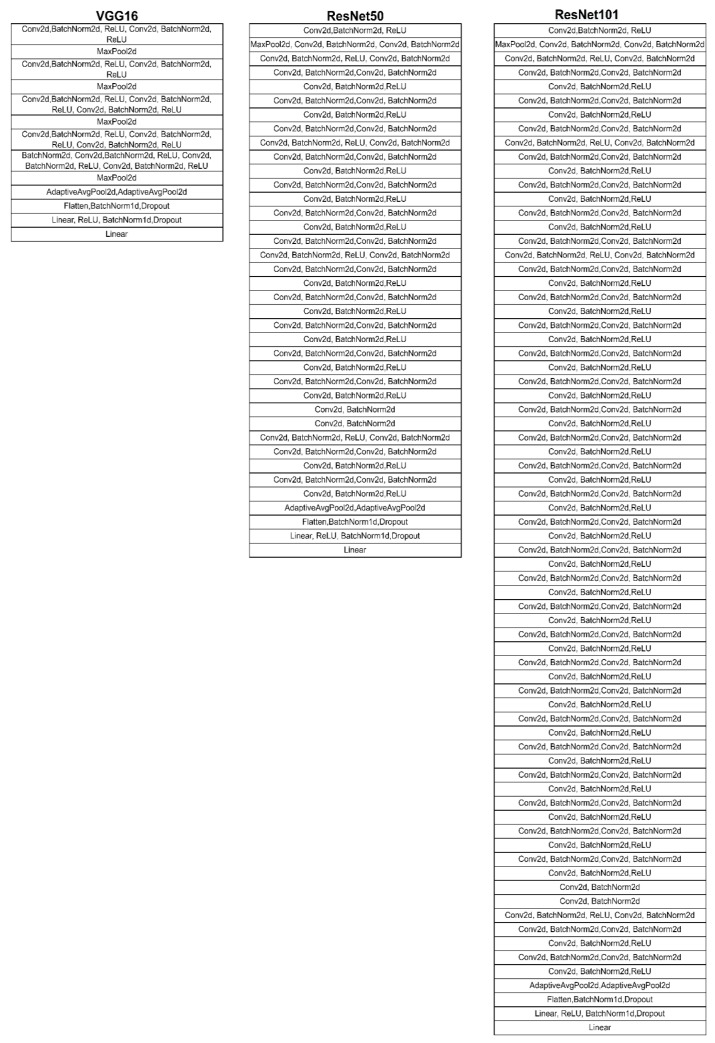
Transfer learning architectures.

**Figure 7 diagnostics-12-03145-f007:**
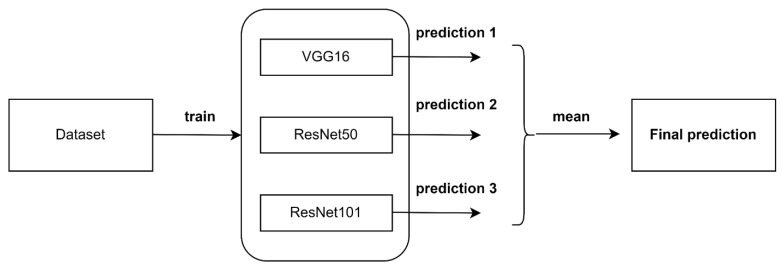
Ensemble algorithm.

**Figure 8 diagnostics-12-03145-f008:**
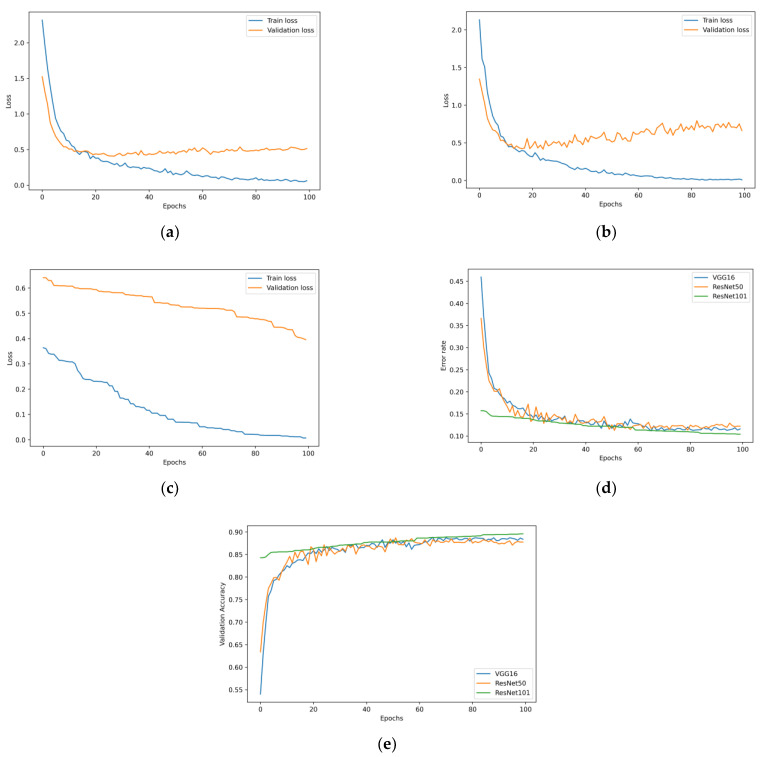
(**a**) Loss plot for VGG16; (**b**) loss plot for ResNet50; (**c**) loss plot for ResNet101; (**d**) error rate; and (**e**) validation accuracy of the three models.

**Figure 9 diagnostics-12-03145-f009:**
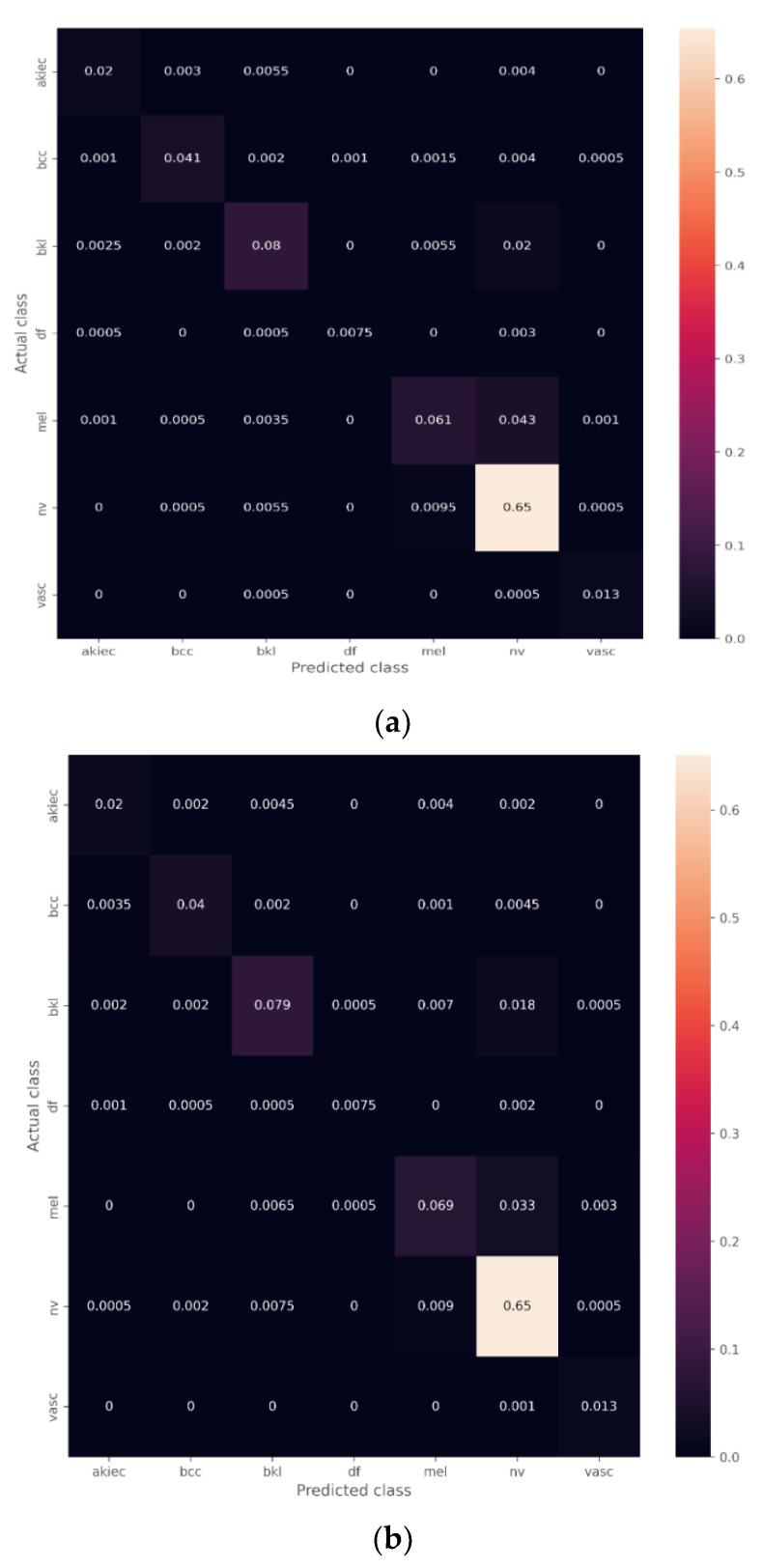
Confusion matrices. (**a**) VGG16; (**b**) ResNet50; (**c**) ResNet101.

**Figure 10 diagnostics-12-03145-f010:**
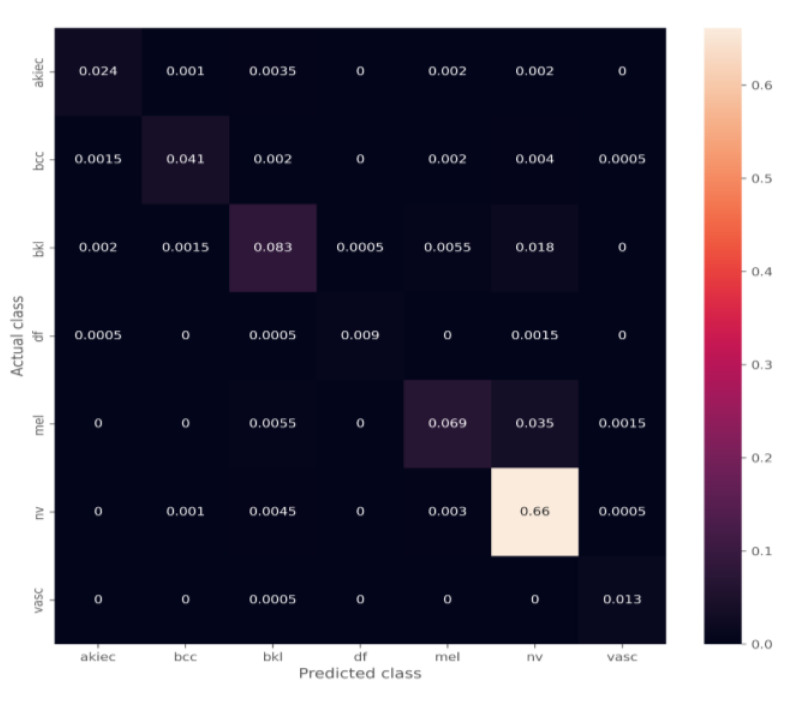
Confusion matrices of the ensemble model.

**Table 1 diagnostics-12-03145-t001:** Dataset statistics.

Class	Train	Validation	Test	Total	Benign/Malignant
Actinic Keratosis (AKIEC)	236	26	65	327	Benign or Malignant
Basal Cell Carcinoma (BCC)	371	41	102	514	Malignant
Benign Keratosis (BKL)	792	88	219	1099	Benign
Dermatofibroma (DF)	83	9	23	115	Benign
Melanoma (MEL)	802	89	222	1113	Malignant
Nevus (NV)	4828	536	1341	6705	Benign
Vascular Cancer (VASC)	103	11	28	142	Benign or Malignant

**Table 2 diagnostics-12-03145-t002:** Image augmentation techniques.

flip	50% of horizontal and vertical flip on all images.
Affine	Translation: Move each image −20 to +20% per axis. Rotation: Rotate each image by −30 to 30 degrees. Scaling: Zoom in each image by 0.5 to 1.5 times.
Multiply	Multiplication of each image by a random value sampled from [0.8, 1.2].
Linear contrast	Change contrast by equation. 127 + alpha × (v-127). V: Pixel value. Alpha: Samples from [0.6, 1.4].
Gaussian Blur	Blur the images using Gaussian kernel with standard deviation sampled from the interval [0.0, 3.0].

**Table 3 diagnostics-12-03145-t003:** Performance evaluation metrics of the three models.

Model	Accuracy (%)	Recall (%)	Precision (%)	F1 Score (%)
VGG16	87.7	75.08	84.66	79.06
ResNet50	87.9	75.57	81.69	78.01
ResNet101	88.15	75.96	84.48	79.57
Ensemble model	90	80.66	88.06	83.77

**Table 4 diagnostics-12-03145-t004:** Class-wise performance of the ensemble models on balanced and unbalanced datasets.

Ensemble Models	Unbalanced Dataset			Balanced Dataset		
Class of Skin Cancer	Recall (%)	Precision (%)	F1 Score (%)	Recall (%)	Precision (%)	F1 Score (%)
AKIEC	73.84	85.71	79.33	84.61	94.82	89.43
BCC	80.39	92.13	85.86	90.19	94.84	92.46
BKL	75.34	83.33	79.13	84.93	90.73	87.73
DF	78.26	94.73	85.71	95.65	95.65	95.65
MEL	61.71	84.56	71.35	72.07	92.48	81.01
NV	98.65	91.62	95.00	99.03	93.91	96.40
VASC	96.42	84.37	90	96.42	90	93.10

**Table 5 diagnostics-12-03145-t005:** Performance evaluation metrics of the three models on the augmented dataset.

Model	Accuracy (%)	Recall (%)	Precision (%)	F1 Score (%)
VGG16	92	85.07	91.84	88.07
ResNet50	92.1	84.65	88.65	86.26
ResNet101	92.25	85.40	90.63	87.79
Ensemble model	93.5	88.98	93.20	90.82

**Table 6 diagnostics-12-03145-t006:** Comparative analysis.

S. No.	Year	Dataset	References	Models	Accuracy	Precision	Recall	F1 Score
1.	2020	HAM10000	[54]	Darknet-53 + NasNet Mobile	83.1%	92.61%	-	88.03%
2.	2019	HAM10000	[55]	ResNet50 + ResNet101	89.8%	-	-	-
3.	2019	HAM10000	[56]	MobileNet	83.23%	-	-	-
4.	2021	KCGMH and HAM10000	[57]	VGG16	85.8%	-	-	-
5.	2021	HAM10000	[58]		90.67%	91.2%	90.3%	89.99%
6.		HAM10000	Proposed work (augmentation + ensemble model)	VGG16 + ResNet50, + ResNet101	93.5%	93.20%	88.98%	90.82%

## Data Availability

The dataset was obtained from [43].

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
