# Peer review of "An Ensemble of Transfer Learning Models for the Prediction of Skin Cancers with Conditional Generative Adversarial Networks"

_diagnostics, 2022, doi:10.3390/diagnostics12123145_

Round 1
Reviewer 1 Report
The paper presents an interesting and important research topic, however, I have the following comments and concerns.
1. The novelty of the paper is limited and needs further clarification and improvement.
2. The detail about the preprocessing steps is missing, authors should clarify whether they have used images in their original form or have applied some preprocessing techniques before passing them to the model for the training.
3. Input images with melanoma are so small just 256x256 pixels. Authors should explain what kind of impact different image sizes will have on the final results. An ablation study will be helpful for this purpose.
4. Figure 2 should show the type of augmentation applied to each image.
5. Skin lesion is a generic term instead of that skin cancer should be preferred when referring to cancerous skin.
6. The paper lacks references to the relevant and latest studies.
7. No explanation is provided about the total number of images of each class after traditional augmentation and data generated by GANS. Authors should add these details in the form of a table.
8. A detailed explanation regarding hyper-parameters and the tuning process and the experimental setup should be provided.
Author Response
An Ensemble of Transfer Learning Models for The Prediction of Skin Cancers with Conditional Generative Adversarial Networks
The authors would like to thank the reviewers for their valuable comments. Authors have responded to all the comments.
Reviewer 1
The paper presents an interesting and important research topic, however, I have the following comments and concerns.
Comment 1. The novelty of the paper is limited and needs further clarification and improvement.
Author Response: The contributions of the work are:
- Explored image augmentation methods such as flip, affine, linear contrast, multiply, and gaussian blur (image transformation methods) to balance the dataset
- Explored the Conditional GAN architecture for generating skin cancer images
- Performance analysis of the fine-tuned pre-trained models VGG16, Res-Net50, and ResNet101 on both balanced and unbalanced datasets.
- An ensemble algorithm by combining the predictions of the three fine-tuned models to improve the performance obtained by deep individual models.
Comment 2. The detail about the preprocessing steps is missing, authors should clarify whether they have used images in their original form or have applied some preprocessing techniques before passing them to the model for the training.
Author Response: Data augmentation is done before passing the images to the model. The following are the details.
- Explored image augmentation methods such as flip, affine, linear contrast, multiply, and gaussian blur (image transformation methods) to balance the dataset
To solve this data scarcity for some classes that will affect the classification model's efficiency, the data augmentation method was employed. Data augmentation is a method that undergoes random transformations on the images to increase the count of images for underrepresented classes without the overhead of collecting more images [33]. The possibilities for image augmentation were enormous such as rotation, translation, and flipping.
In the present work, we did several operations as shown in Table 2. All of them were available on the python library imaging [34]. The images after augmentation are shown in Fig 2.
4.2 Transfer learning model on the balanced data obtained by data augmentation
Several image augmentation techniques were used to increase the number of images within the six lesions. The skin lesion type "NV" images were excluded from the data augmentation process since it was an overrepresented class. Only underrepresented groups underwent the image augmentation procedure. In addition, we attempted to create synthetic images for each class. However, as shown in Fig. 4, the produced images did not appear to possess the characteristics that differentiate the seven types of skin lesions. We anticipate that such data will not be suitable for training the model for optimal performance. Therefore, the resulting CGAN data was excluded from the training data with augmentation. The performance of the developed models VGG16, ResNet50, and the ResNet101 are shown in Table 5.
Data augmentation was the most straightforward response to this problem. However, the data augmentation might be one of two different types: picture transformations using the conventional methods, or the development of synthetic images using the capabilities of GAN architecture. Several of the earlier researchers successfully implemented the GAN algorithm as a means of image augmentation, and they obtained performance improvements as a result [43][29].
Comment 3. Input images with melanoma are so small just 256x256 pixels. Authors should explain what kind of impact different image sizes will have on the final results. An ablation study will be helpful for this purpose.
Author Response: We are using the latest developments in deep learning i.e., transfer learning (TL). TL used various pre-trained models based on ImageNet dataset. These models have been developed with a exact restriction of the image size i.e., 256*256. Hence, we are unable to verify the effect of images with various file sizes
Comment 4. Figure 2 should show the type of augmentation applied to each image.
Author Response:
The following techniques were applied to the images (a) AKIEC (b) BCC (c) BKL (d) DF (e) MEL and (f) VASC
Table 2. Image augmentation techniques
flip |
50% of horizontal and vertical Flip on all images. |
Affine |
Translation: move each image -20 to +20% per axis Rotation: Rotate each image by -30 to 30 degrees Scaling: Zoom in each image by 0.5 to 1.5 times |
Multiply |
Multiplication of each image by a random value sampled from [0.8,1.2]. |
Linear contrast |
Change contrast by equation 127+alpha*(v-127) V: pixel value Alpha: samples from [0.6,1.4] |
GaussianBlur |
Blur the images using Gaussian kernel with standard deviation sampled from the interval [0.0,3.0]. |
Figure 2. Augmented images(a) AKIEC (b) BCC (c) BKL (d) DF (e) MEL and (f) VASC
Comment 5. Skin lesion is a generic term instead of that skin cancer should be preferred when referring to cancerous skin.
Author Response: We found 66 places were lesion was written. Now it is changed to cancer in all the places.
Comment 6. The paper lacks references to the relevant and latest studies.
Author Response: The following papers are added in the reference list.
- Mohakud, Rasmiranjan, and Rajashree Dash. "Designing a grey wolf optimization based hyper-parameter optimized convolutional neural network classifier for skin cancer detection." Journal of King Saud University-Computer and Information Sciences 34, no. 8 (2022): 6280-6291.
- Shorfuzzaman, Mohammad. "An explainable stacked ensemble of deep learning models for improved melanoma skin cancer detection." Multimedia Systems 28, no. 4 (2022): 1309-1323.
- Gouda, Walaa, Najm Us Sama, Ghada Al-Waakid, Mamoona Humayun, and Noor Zaman Jhanjhi. "Detection of Skin Cancer Based on Skin Lesion Images Using Deep Learning." In Healthcare, vol. 10, no. 7, p. 1183. MDPI, 2022.
- Tajjour, Salwan, Sonia Garg, Shyam Singh Chandel, and Diksha Sharma. "A novel hybrid artificial neural network technique for the early skin cancer diagnosis using color space conversions of original images." International Journal of Imaging Systems and Technology (2022).
- Kousis, Ioannis, Isidoros Perikos, Ioannis Hatzilygeroudis, and Maria Virvou. "Deep Learning Methods for Accurate Skin Cancer Recognition and Mobile Application." Electronics 11, no. 9 (2022): 1294.
- Rajput, Gunjan, Shashank Agrawal, Gopal Raut, and Santosh Kumar Vishvakarma. "An accurate and noninvasive skin cancer screening based on imaging technique." International Journal of Imaging Systems and Technology 32, no. 1 (2022): 354-368.
- Huynh, Anh T., Van-Dung Hoang, Sang Vu, Trong T. Le, and Hien D. Nguyen. "Skin Cancer Classification Using Different Backbones of Convolutional Neural Networks." In International Conference on Industrial, Engineering and Other Applications of Applied Intelligent Systems, pp. 160-172. Springer, Cham, 2022.
Comment 7. A detailed explanation regarding hyper-parameters and the tuning process and the experimental setup should be provided.
Author Response:
3.5 Experimental Setup
The quantity of hyper-parameters rises as CNN depth increases. The CNN model has a large number of hyper-parameters, including the number of layers, the number of kernels in each layer, the size of the dense layer, the learning rate, the number of convolutions, the number of max-pooling, the number of dense layers, the number of epochs, the batch size, the optimizer, the activation function, and many others. The selection of these hyper-parameters is typically done manually and involves expensive trial and error. As a result, CNN training typically takes a long time because different hyper-parameter configurations are evaluated in different rounds. In the interim, new CNNs typically have an accelerated increase in the number of layers together with hyper-parameters. Therefore, trying to manually determine the almost ideal hyper-parameter setup for a CNN at a reasonable cost is virtually pointless. On the other hand, effective hyper-parameter adjustment can improve the CNN model's overall performance. Because of this goal, many researchers have turned to the tuning of CNN hyper-parameters as an optimization problem. These hyper-parameters have a significant impact on the CNN model's result. The near-optimal hyper-parameter setup for a CNN cannot be manually determined by cost-effectively examining all conceivable combinations. Therefore, correct hyper-parameter tuning of CNN is designed as an optimization problem with the goal of improving the CNN model's overall performance.
There are 16 layers in VGG16 CNN. The input dimension to the network is (224,224,3). What makes VGG16 stand out from other implementations because it uses a 2x2 stride 2 filter with the same padding and maxpool layer, instead of having many hyper-parameters. Throughout the architecture, the convolution and max pool layers are arranged in the same manner. For output, there are two Fully Connected (FC) and a softmax layer.
The 16-layer network used by VGG16, which was introduced in 2014 by Simonyan and Ziserman of the Visual Geometry Group Laboratory at Oxford University11, is much deeper than AlexNet but provides a simpler network by substituting numerous 3 3 kernel-sized filters for a single large kernel-sized filter. ImageNet, a collection of more than 14 million photos divided into 1000 groups, scored 92.7% on the top-5 test. Figure 3 depicts the VGG16 architecture.
The ReLU from AlexNet is carried by the 13 convolutional layers and 3 fully linked layers of the VGG16. The fully connected layers are utilized for classification, whereas the convolutional layers are used to extract features. Each of the 64 feature kernel filters in the first and second convolutional layers is 3 * 3 in size. As the input image (a RGB image with depth of 3) moves through the first and second convolutional layers, its dimensions are modified to 224 * 224 * 64. The output is subsequently transferred with a stride of 2 to the max pooling layer. With a filter size of 3 * 3, the third and fourth convolutional layers are composed of 128-feature kernel filters. After these first two layers are applied, a max pooling layer with stride 2 is used to compress the output to 56 * 56 * 128 pixels. The fifth, sixth, and seventh layers are convolutional layers with a kernel size of 3 * 3. All three of these layers combine to employ 256 feature maps. A max pooling layer with stride 2 is placed after these layers. With kernel sizes of 3*3, there are two sets of convolutional layers from the eighth to the thirteenth. Each of these convolutional layers has 512 kernel filters. A max pooling layer with a stride of 2 follows these layers. The 16th layer, which has 1000 neurons, is a softmax output layer for classifying 1000 photos from the ImageNet data set. The 14th and 15th layers are fully linked hidden layers with 4096 neurons each. VGG16 are utilized as feature extractors and classifiers in image processing applications notwithstanding its utility in image processing and classification.
Name Specifications
GPU GPU DELL EMC 740
RAM 128 GB
GPU RAM 32 GB
Language Python
IDE Jupyter notebook environment on Google Colabs
Therefore, this combination of algorithm hyperparameters (i.e., Optimizer of sgdm, Mini BatchSize of 40, MaxEpochs of 5, and Initial LearnRate of 10–4) was selected for Resnet50 and Resnet101 transfer learning

Reviewer 2 Report
In this study, Amal Al-Rasheed at al. fit an ensemble of deep learning models for classify skin lesions with conditional GAN.
Here are my comments:
1) is it possible to try other better CNN models, e.g. efficientNet;
2) how the authors handle the imbalance of the multi-class samples, even though they tried augmentation; and any better methods?
3) they used the mean to process the predictions of the three CNN as the ensemble algorithm, is there any other algorithm? e.g. logistic regression, tree-based algorithm?
4) a graphic abstract or a detailed flowchart are highly recommended.
Author Response
An Ensemble of Transfer Learning Models for The Prediction of Skin Cancers with Conditional Generative Adversarial Networks
The authors would like to thank the reviewers for their valuable comments. Authors have responded to all the comments.
Reviewer 2
In this study, Amal Al-Rasheed at al. fit an ensemble of deep learning models for classify skin lesions with conditional GAN.
Here are my comments:
Comment 1: is it possible to try other better CNN models, e.g. efficientNet;
Author Response: In this work we have implemented three pre-trained model which already gave good results compared to other works. As suggested we can also try EfficientNet in the future work. This technique is already applied by few researchers in handling different problems.
Comment 2: how the authors handle the imbalance of the multi-class samples, even though they tried augmentation; and any better methods?
Author Response: In addition, we investigated the idea of generating synthetic data to solve the class imbalance issue. The well-known CGAN [35] architecture is used to create the images. Fig. 3 depicts the high-level design of the network.
As a consequence of this, the majority of the medical datasets that are accessible to the public have a problem with data imbalance [42]. As can be observed in Table 1, the dataset that was used for this investigation has some serious imbalances.
The currently developed model could achieve greater results, but the dataset suffered from an imbalanced data problem, which had a direct impact on the performance of the model developed from the dataset. With the typical data augmentation technique, performance has been enhanced.
On the other hand, the image transformations might be able to produce a magnificent dataset to solve the problem of image imbalance. In addition, an ensemble model that consisted of the VGG16, ResNet50, and ResNet101 models that had been trained on both balanced and unbalanced datasets was developed, and its performance was analyzed.
Comment 3: they used the mean to process the predictions of the three CNN as the ensemble algorithm, is there any other algorithm? e.g. logistic regression, tree-based algorithm?
Author Response: Yes, we have implemented three pre-trained model and calculated the accuracy based on the mean. Infact, as suggested we can also use other algorithms like logistic regression and tree-based algorithms, but these are not new and it can be tried as a future direction and see how the results show. Infact, it is very interesting to use these algorithms for detecting the skin cancer.
Comment 4: a graphic abstract or a detailed flowchart are highly recommended.
Author Response:

Round 2
Reviewer 1 Report
Authors have addressed most of my concerns satisfactorily well.
Author Response
Thank you very much.

Reviewer 2 Report
a graphic abstract or a detailed flowchart are still highly recommended.
Author Response
Author Response: A new figure is included in the paper.
Figure 1. The complete flow of the work.

Round 3
Reviewer 2 Report
No more comment